# Advancements in the Application of scRNA-Seq in Breast Research: A Review

**DOI:** 10.3390/ijms252413706

**Published:** 2024-12-22

**Authors:** Zhenyu Zhang, Xiaoming Ma, Yongfu La, Xian Guo, Min Chu, Pengjia Bao, Ping Yan, Xiaoyun Wu, Chunnian Liang

**Affiliations:** 1College of Animal Science and Technology, Gansu Agricultural University, Lanzhou 730070, China; zzy960911@163.com; 2Key Laboratory for Yak Genetics, Breeding, and Reproduction Engineering of Gansu Province, Gansu Provincial Key Laboratory of Yak Breeding Engineering, Lanzhou Institute of Animal Husbandry and Veterinary Medicine, Chinese Academy of Agricultural Sciences, Lanzhou 730070, China; maxiaoming@caas.cn (X.M.); layongfu@caas.cn (Y.L.); guoxian@caas.cn (X.G.); chumin@caas.cn (M.C.); baopengjia@caas.cn (P.B.); yanping@cass.cn (P.Y.); wuxiaoyun@caas.cn (X.W.); 3Key Laboratory of Animal Genetics and Breeding on Tibetan Plateau, Ministry of Agriculture and Rural Affairs, Lanzhou 730070, China

**Keywords:** scRNA-seq, breast, single cell, advancements

## Abstract

Single-cell sequencing technology provides apparent advantages in cell population heterogeneity, allowing individuals to better comprehend tissues and organs. Sequencing technology is currently moving beyond the standard transcriptome to the single-cell level, which is likely to bring new insights into the function of breast cells. In this study, we examine the primary cell types involved in breast development, as well as achievements in the study of scRNA-seq in the microenvironment, stressing the finding of novel cell subsets using single-cell approaches and analyzing the problems and solutions to scRNA-seq. Furthermore, we are excited about the field’s promising future.

## 1. Introduction

The breast is a vital organ for producing milk and raising the young. The parenchyma and stroma make up the tissue of the breast. The stroma is made up of lymphatic ducts, nerves, connective tissues, and adipose pads. The adipose pads are primarily responsible for supporting the growth and development of the breast parenchyma, acting as a site for the differentiation of milk ducts, and controlling the synthesis of growth factors. The breast parenchyma, the primary site for milk synthesis and secretion, is composed of alveoli, which are the basic units for milk secretion and synthesis. In the dairy animal models, these alveoli are aggregated by numerous lobules, and each lobule is constituted by a single layer of breast epithelial cells [1]. The inside of the alveoli is lined with a single layer of breast epithelial cells, while the outside is surrounded by myoepithelial cells. Mature breast possess the function of secreting milk, which is then transported by breast ducts synthesized and secreted by epithelial cells. Depending on age and reproductive status, the breast undergo dynamic cyclical processes of growth, differentiation, and regression [2].

The breast undergoes a developmental process starting from embryogenesis, where the ectoderm on the embryo induces the formation of breast buds. These buds further develop and differentiate into milk vesicles and milk ducts [3]. Prior to puberty, breast development focuses on the development of the stroma, including connective and adipose tissues, which provide support for subsequent breast development. During this stage, the growth rate of the breast is synchronized with the overall growth of the body [4].

When entering puberty, with the increase in hormone secretion, the breast enters a stage of rapid development, often exceeding the normal growth rate of the body. The breast ducts extend rapidly into the adipose tissue and form numerous branches, constituting a complex system of breast ducts, accompanied by the emergence of terminal buds and an increase in the size of the breast. However, at this stage, the breast alveoli are not yet fully matured [5]. During pregnancy, early breast development continues to accelerate under the influence of hormones such as progesterone and estrogen. The number of breast ducts continues to increase as the third ductal branch forms. The ends of these branches begin to develop into alveoli, but these initial alveoli are unable to secrete milk [6]. In mid-gestation, secretory lumens appear in the follicles, gradually expanding and replacing fat and connective tissue [7]. In late pregnancy, breast cells differentiate into highly secretory unilamellar cuboidal and columnar epithelial cells that acquire the capacity to produce milk [2]. During lactation, the breast cells are highly active and enter a lactation peak. After this, the breast undergo degeneration and the lobules stop producing milk in the later stages of lactation. As the internalization process proceeds, the alveoli atrophy and the interstitium increase, gradually replacing the alveoli. Breast tissue degenerates and milk synthesis and secretion cease [8].

The secretory cells of the breast tissue, with nutrients in the blood as raw materials, produce milk in the cells, and then secrete it into the alveolar lumen in a process called lactation; the milk in the alveolar lumen, through the ductal system of the breast tissue, converge step by step, and then finally flow to the outside of the body through the breast ducts and nipple ducts, in a process known as excretion of breast milk. The two processes of milk secretion and lactation, which are different in nature but interrelated, are collectively called lactation [9]. This process of milk synthesis and secretion is called lactation, and it lasts from the beginning of labor until the next birth, constituting a lactation cycle [10]. Throughout the cycle, breast epithelial cells undergo proliferation and apoptosis. Self-renewal and apoptosis of breast epithelial cells are essential for sustained lactation [11]. Once lactation begins, the breast continues this activity for an extended period of time. The number of breast epithelial cells with milk-producing functions and their secretory capacity significantly affect the breast’s ability to produce milk, the number of breast epithelial cells is closely related to lactation [12]. When the breast degenerates into the dry milk phase, this self-renewal phase is essential for the subsequent lactation cycle [13]. During this stage, the proliferative and differentiating capacity of breast epithelial cells gradually increases, reaching a peak around the time of delivery, when the number of breast epithelial cells is highest and breast development reaches its highest level. After delivery, the proliferative capacity of breast epithelial cells decreases significantly, but a significant proportion of cells remain self-renewing, helping to maintain the total number of breast cells.

Previous transcriptome sequencing methods have typically treated entire organs or tissues as homogeneous, resulting in average gene expression reads that ignore heterogeneity among individual cells [14]. This approach presents challenges in analyzing tiny tissues and single cells [15]. The development of single-cell sequencing technology provides opportunities to study rare tissues and individual cell characteristics. Currently, single-cell RNA sequencing (scRNA-seq) is widely used in stem cell differentiation, tumor research, exploration of disease subtypes, organ development, and other related areas of research [16]. The breast consists of many different types of cells, and the development of the different types of cells is not synchronized [17]. The construction of single-cell maps based on scRNA-seq technology allows for the clear characterization of complex cell types within breast tissue, thus exploring gene regulatory mechanisms in specific cell development at the single-cell level [18]. In this paper, we review the cell types involved in breast development, and the research progress of scRNA-seq technology in breast research. In addition, we provide new insights into the future trends of scRNA-seq.

## 2. Different Cell Types in Breast

### 2.1. Luminal Hormone-Responsive Epithelial Cells

Luminal hormone-responsive epithelial cells are a specialized group of cells in the breast that are located in the lumen of the milk ducts and react primarily with hormones [19]. Hormone-responsive epithelial cells are sensitive to hormones such as estrogen and progesterone, and the hormones trigger intracellular signaling pathways that lead to cell proliferation and differentiation by binding to cell surface receptors [20]. During the female physiological cycle, the number and activity of these cells change in response to hormone levels as part of the normal physiological functioning of the breast. In certain pathologies, such as breast hyperplasia and breast cancer, hormone-responsive epithelial cells in the lumen show abnormal morphology, proliferation rate and function [21]. The abnormalities may be related to receptor mutations, signaling pathway abnormalities or genetic background. Hormone-responsive epithelial cells, as an important part of the ductal lumen of the breast, maintain the normal morphological structure of the breast together with other surrounding cells and tissues [20]. These cells are involved in the secretion of breast tissue fluid, provide support for the lactation process, and respond to hormones that adapt the breast to changes in the physiological cycle, playing a key role in breast reproduction.

### 2.2. Luminal Secretory Epithelial Cells

Luminal secretory epithelial cells, a specialized cell type in the breast, play an important role in both physiological and pathological conditions [22]. These cells communicate material and information with the external environment through specialized structures on the cell membrane, microvilli and secretory vesicles. Prolactin and progesterone have important stimulatory and regulatory effects on the luminal secretory epithelial cells of the breast, and together they promote breast development and lactation. The synergistic action of these hormones ensures the normal physiological function of breast tissue [23]. Hormones regulate cellular secretion by binding to cell surface receptors and triggering intracellular signaling pathways. In addition, the luminal secretory epithelial cells are also involved in milk secretion, and under the stimulation of hormones, they promote the synthesis and secretion of various components of milk, and through a series of biochemical reactions and substance transportation, they transport nutrients and active substances into the lumen of the breast ducts to eventually form milk [24]. This process is supportive of neonatal development and homeostasis within the breast. In addition, during puberty and pregnancy, the cells proliferate and differentiate in response to hormones and form the ductal lumens and lobules of the breast ducts, the formation of which is necessary for the maintenance of normal breast function and lactation [25]. Luminal secretory epithelial cells are also implicated in breast disease, and usually the cells show abnormal proliferation, differentiation, or dysfunction [26]. Intensive study of such cells could further reveal the mechanisms of breast diseases and lead to new therapeutic approaches.

### 2.3. Basal Myoepithelial Cells

The basal myoepithelial cells of the breast are a type of cell located in the basal membrane of the breast tissue [27]. These cells are flat or spindle shaped and form continuous layers through tight connections [28]. Basal myoepithelial cells play a key role in the breast [29]. These cells act as a support structure, regulating breast tissue tension through contraction and expansion, maintaining breast shape and stability, and enabling the breast to perform its normal physiological functions. Basal epithelial cells also participate in immune defense, recognizing and eliminating potential pathogens and protecting the breast from invasion [30]. Basal epithelial cells also secrete biologically active substances, such as growth factors and chemokines, which play a regulatory role in breast growth, differentiation and repair. In addition, these cells interact with other cells to maintain breast homeostasis [31]. Basal myoepithelial cells are also involved in breast development and participate in breast formation and remodeling by regulating their own proliferation, differentiation, and apoptosis during all stages of development. Another characteristic of basal myoepithelial cells is their ability to migrate, and when the breast is damaged or inflamed, they migrate to the damaged area and participate in repair. At the same time, they also have the signal transduction ability to receive and transmit signals from other cells or tissues to participate in the physiological and pathological regulation of the breast, and such signals make the basal myoepithelial cells a center of information on breast tissue.

### 2.4. Endothelial Cells

The breast endothelial cells are located in the inner wall of the blood vessels of the breast. Endothelial cells are flexible and plastic, with tight connections between cells to accommodate morphological changes in blood vessels and lymphatic vessels [32]. In the breast vasculature, endothelial cells are categorized according to the location and morphology of the cells, and they are divided into arterial endothelial cells and venous endothelial cells. Arterial endothelial cells are thicker and can withstand higher blood pressure, while venous endothelial cells are thinner and adapt to lower blood pressure [32]. Additionally, lymphatic endothelial cells present in the lymphatic vessels of the breast are responsible for lymphatic transport and immunity [33]. Endothelial cells play a variety of important roles in breast physiology, they form the inner structure of blood vessels and lymphatic vessels, maintain the integrity and permeability of blood vessels, participate in the regulation of vasodilatation and contraction, and influence the flow of blood and lymphatic vessels. Additionally, breast endothelial cells have certain metabolic functions and are involved in the exchange and transportation of nutrients and metabolites. The endothelial cells of the breast also respond to hormones, and estrogen and progesterone bind to receptors on the endothelial cells, and this hormonal response regulation mechanism plays a key role in physiologic processes such as breast development, the menstrual cycle, pregnancy, and breastfeeding [34]. The breast endothelial cells interact with other cells through cell junctions and signaling [35]. For instance, it interacts with immune cells involved in the immune and inflammatory response in the breast. Endothelial cells communicate with epithelial and stromal cells, essential for maintaining breast growth and development. In immunity, endothelial cells recognize and respond to foreign pathogens, activate and eliminate harmful substances and secrete immune-associated factors, such as inflammatory mediators and chemokines, which are integral to breast immunity and repair.

### 2.5. Immune Cells

The breast immune cells have the function of maintaining breast immune homeostasis, defending against pathogens, and participating in breast repair and regeneration [36]. There are several types of immune cells in the breast, each with specific characteristics and functions [37]. The breast immune cells include macrophages, dendritic cells, T cells, B cells and NK cells [38]. Macrophages are responsible for phagocytosis and elimination of pathogens, apoptotic cells, and regulation of the immune response [39]. Dendritic cells, which present antigens, recognize and capture invading pathogens, present them to T cells to initiate an immune response [40]. T cells are involved in activating and suppressing immune cells, including anti-tumor immune responses [41]. B cells play a role in antibody production, conducting immune responses against specific pathogens and providing immune memory [42]. NK cells directly attack and destroy infection or tumor cells and are natural immune defense mechanisms [43]. In addition, breast immune cells are also involved in tumor development, recognition and attack, anti-tumor immunity, and inhibition of tumor growth and spread [44]. In-depth study of breast immune cells is helpful to better understand the immune mechanism and provide ideas for the prevention and treatment of breast diseases.

### 2.6. Mesenchymal Cells

The breast mesenchymal cells (MSCs) are composed of non-epithelial components and exist in the mesenchyma, providing structural support and microenvironmental regulation for the breast. The cytoplasm contains organelles and fibers that form complex cell networks and secrete growth factors [45], matrix proteins and other factors, breast mesenchymal cells maintain breast structural and functional homeostasis [46]. MSCs are categorized into different subtypes based on origin, phenotype and function. These cells differ in differentiation potential and surface markers, such as adipose-derived MSCs [47] and bone marrow mesenchymal stem cells [48]. These cells have the potential to self-renew and differentiate into various cell types that play an important role in regeneration and repair.

Breast MSCs play a key role in breast development, physiology and disease, secreting growth factors, chemokines and matrix proteins that regulate the proliferation, differentiation and migration of epithelial cells and participate in tissue formation and maintenance. In addition, MSCs interact with immune cells to regulate breast immune and inflammatory responses [49]. MSCs are an important component of the tumor microenvironment in the development of breast cancer [50]. MSCs accelerate cancer cell proliferation, migration and invasion by secreting tumor growth factors and angiogenic factors. Breast MSCs influence breast cancer progression and metastasis by regulating immunity and inflammation. Breast mesenchymal stromal cells also have significant immunomodulatory capacity and play a role in breast cancer immune homeostasis and immune escape by secreting immunomodulatory factors, anti-inflammatory factors, or immunosuppressive molecules that regulate the activity and function of immune cells such as T, B, and NK cells [51]. Breast mesenchymal stromal cells also promote angiogenesis, which provides nutrients and oxygen to cancer cells and promotes tumor growth and metastasis [52]. They secrete vascular endothelial growth factor, which promotes blood vessel formation and expansion in breast cancer tissues. Due to their function, breast MSCs have become a target for breast cancer treatment. By utilizing the targeting ability of MSCs, drugs can be delivered precisely to the tissues, improving treatment efficacy and reducing side effects [53]. Additionally, breast mesenchymal cells also serve as potential biomarkers for breast cancer diagnosis and prognostic assessment [54].

### 2.7. Perivascular Cells

The perivascular cells are often referred to vascular smooth muscle cells [55]. Together with endothelial cells, it forms the basic structure of the blood vessel wall [56]. These perivascular cells contain large amounts of contractile proteins that regulate the contraction and dilation of blood vessels.

Perivascular cells are closely related to endothelial cells and maintain normal physiological functions by secreting growth factors and signals. In addition, perivascular cells monitor endothelial cell health and quickly initiate repair when damage or abnormalities are detected [57]. Perivascular cells are closely associated with endothelial cells and maintain normal physiological function by secreting growth factors and signals. In addition, perivascular cells monitor endothelial cell health and rapidly initiate repair when damage or abnormalities are detected. When blood flow increases, the cells release relaxing factors to dilate the blood vessels and, conversely, in the presence of decreased blood flow, contracting factors are released to constrict the blood vessels and reduce blood flow. This regulation ensures adequate blood supply to the breast tissue under different physiological conditions, and the perivascular cells also have the ability to remove cellular debris from the vessel wall and phagocytosis [58]. They recognize and phagocytose apoptotic endothelial cells, platelets, and other cellular debris, preventing accumulation on the vessel walls. Scavenging and phagocytosis help to keep blood vessels clean and healthy, and although they are not directly involved in the construction of the blood–brain barrier, they exhibit regulatory mechanisms similar to those of the cells surrounding the blood–brain barrier that affect vascular permeability [59]. Perivascular cells also have the ability to proliferate and differentiate. Under vascular injury or inflammation, neoplastic cells rapidly proliferate and differentiate and participate in the repair and reconstruction of the vessel wall [60]. In some cases, they differentiate into other cells, such as smooth muscle or endothelial cells, to meet physiological needs [61]. In-depth study of breast perivascular cells can improve our understanding of breast physiology and pathology, and provide insights for the prevention and treatment of breast related diseases.

### 2.8. Adipocyte and Mast Cells

Adipocytes in the breast are found mainly in the interstitium and are characterized by being rich in lipid droplets [62]. The main function of fat cells in the breast is to store fat and provide energy reserves for the breast [63].

Mast cells are also an important cell type widely distributed in various organ tissues with a distinctive granular structure containing biologically active mediators such as histamine and heparin [64]. When cells are stimulated, mediators are rapidly released, triggering a series of inflammatory and immune responses [65]. Furthermore, mast cells are involved in the remodeling and repair of breast [66].

## 3. scRNA-Seq Technology and Its Application in Breast

Single-cell RNA sequencing (scRNA-seq) is a new technique for unbiased, high-throughput, high-resolution transcriptome analysis at the single-cell level [67]. While traditional transcriptome sequencing can only provide the average transcript levels of all cells in a heterogeneous sample, scRNA-seq can accurately characterize the transcripts of each cell in the sample [68]. This technology could lead to further understanding of the intrinsic histological differences between similar cells, exploration of known cell type transcriptome signatures, and the discovery of new cell types.

In 2009, Tang [69] used single-cell mRNA-seq for the first time to analyze the ovoid globules of a single mouse, bringing a new breakthrough in the field of transcriptome analysis and demonstrating the great potential and wide application prospects of single-cell mRNA-Seq technology. The first step in scRNA-seq involves the isolation of individual cells, which is traditionally accomplished by extracting cells from a cell suspension under a microscope using a capillary pipette, although this is inefficient [70]. Currently, flow cytometric sorting is the method of choice for isolating high-purity single cells, utilizing fluorescent monoclonal antibodies to identify specific markers on the cell surface and to classify different cell populations [71]. Microfluidics utilizes microscale channels to isolate single cells from microliters to milliliters of samples, and is widely used because of its low cost, low sample consumption, and high separation efficiency [72].

Currently, scRNA-seq is commonly used as a library preparation platform by 10× Genomics, which uses microfluidics to capture individual cells and generate individual libraries for each cell [73]. The specific workflow involves mixing single cells with gel beads in oil droplets and then disrupting the cell membrane to release mRNA, which is then contacted with reverse transcriptase, nucleic acid primers on the gel beads and dNTP substrate to initiate reverse transcription and PCR amplification [74]. The amplified cDNA is fragmented and the resulting DNA fragment consists of a barcode, UMI and PolyT. The barcode is a 16-base sequence unique to each gel bead, and the UMI is a random sequence used to differentiate the DNA fragments. Using 10× Genomics acquires large amounts of cellular data at a low time and cost, giving it a clear advantage in processing large cell populations and detecting rare cell types. Using 10× Genomics, one can acquire large amounts of cellular data in a short time and at a low cost, giving it a clear advantage in processing large cell populations and detecting rare cell types.

Development is driven and controlled by temporal and spatial variations in gene transcription, which translates the resulting messenger RNA (mRNA) into proteins [75]. In recent years, with advances in sequencing technology, substantial progress has been made in the study of the breast using scRNA-seq, but the understanding and definition of the various cell types within breast tissue are still not precise enough [25]. Accurate cell categorization is paramount to the analysis of single-cell sequencing data and involves manual annotation of cell marker genes for different cell types [76]. By accurately identifying cell types, it is possible to validate cell function and differentially express functional genes and construct trajectories of cellular differentiation with specific significance [77]. Currently, we can utilize software to assist in cell type identification, with key websites including CellMarker (http://xteam.xbio.top/CellMarker/) (accessed on 15 October 2024) and SingleR (http://github.com/dviraran/SingleR) (accessed on 16 October 2024). In Table 1, we provide examples of marker genes used as reference breast tissue markers. Some marker genes are not unique to a single cell type, but can be used to identify and annotate different cell types, suggesting similarities between different cell types. Therefore, the process of using marker genes to annotate cells is particularly important in single-cell RNA sequencing, and as many marker genes as possible need to be selected to achieve the desired results.

In recent years, there has been a gradual increase in the number of articles on breast research using scRNA-seq technology, with studies focusing on the study of cells at different developmental stages of the breast. These studies have comprehensively characterized breast cell subpopulations and trajectories of differentiation, and elucidated breast characteristics as well as the relevant molecular mechanisms of regulation at the single-cell level.

Bach [78] revealed for the first time the dynamic differentiation of human breast epithelial cells using scRNA-seq. This study also identified progenitor cell populations capable of differentiating into follicular cells and hormone-sensing progenitors, and their dynamic changes that occur during pregnancy, lactation, and recurrence. This study uses single-cell sequencing technology to reveal the developmental process of the adult breast, which will provide an important reference for subsequent understanding of the relationship between other cell types of the breast and breast cancer development.

Although the adult breast undergoes significant expansion and differentiation during pregnancy and lactation to adapt to milk synthesis and secretion in order to sustain the offspring, little is still known about the mechanisms of human breast tissue remodeling during this physiological process due to difficulties in sample acquisition. To delve deeper into this area, Alecia [79] performed a single-cell transcriptome analysis covering a total of 110,744 surviving breast cells isolated from nine human breast donors and seven non-lactating breast tissue donors, which revealed that human breast milk consists predominantly of epithelial cells of the tubulo-luminal pedigree and a range of immune cells, and further in-depth analyses of the breast cell transcriptomes allowed us to identified two distinct secretory cell types that share similarities with ductal progenitor cells, but did not find a population that directly corresponds to hormone-responsive cells. Taken together, this study not only provides a valuable reference map of cellular dynamics during human lactation, but is also expected to provide new insights into the complex interactions between pregnancy, lactation and breast cancer.

The highly heterogeneous nature of breast epithelial tissue is critical for understanding normal cellular homeostasis and tumorigenesis. Breast epithelial tissue consists of an outer basal layer and an inner luminal layer, which includes multiple cell types. In recent years, the application of single-cell RNA sequencing (scRNA-seq) technology has significantly advanced the resolution of differentiation hierarchies of breast epithelial cells, revealed new cell types and states, and enabled comprehensive unbiased analyses of heterogeneous histiocytes. reviewing these advances, Joseph [80] succeeded in identifying c-Kit+ similar to that in recent scRNA-seq studies progenitor cells and specific cellular states, providing an important basis for a deeper understanding of the cellular heterogeneity of the breast epithelium and its role in physiopathologic processes.

The female breast epithelium undergoes significant remodeling during physiological processes such as development, pregnancy, and menopause, and these changes have been associated with an increased risk of breast cancer. Using single-cell RNA sequencing, Kohei [81] constructed an integrative map encompassing both the mouse (50K) and human (24K) breast epithelium, revealing differentiation trajectories from the origins of the embryonic breast stem cell to the three epithelial lineages (basal cells, tubulo-hormonal-sensing cells, and ductal alveolar cells) differentiation trajectories. Combined with cancer genome mapping, human breast cancer single-cell RNA sequencing information and the association of glandular remodeling with breast cancer subtypes, we further deduced the cell types of breast cancer origins and provided a comprehensive gene expression profile of breast cells, which revealed the effects of internal and external stimuli on the breast epithelium with high precision, and contributed to the in-depth understanding of molecular mechanisms of breast cancer development.

The cells that constitute complex life tissues or organs originate from the differentiation of multipotent stem cells or progenitor cells [82]. Rajshekhar [83] has successfully generated a comprehensive single-cell transcriptome atlas encompassing 6060 individual cells and 22,184 expressed genes by analyzing breast gland tissues at five crucial developmental time points, ranging from embryonic to adulthood in mice. The map reveals the evolution of cellular states at the transcriptional and epigenetic levels during breast development and distinguishes between fetal breast stem cells (fMaSC) and their precursor and progeny cells. fMaSC exhibit unique gene expression patterns involving balanced co-expression of factors associated with the adult lineage and specific metabolic profiles, which are abrogated during maturation but are recapitulated in human breast cancer.

Nguyen [27] identified 25,790 primary human breast epithelial cells in the breast tissue of seven patients who underwent breast reduction surgery using scRNA-seq. Unbiased clustering revealed the presence of three distinct epithelial cell subpopulations: one basal cell type and two luminal cell types, which were further characterized as secretory L1 and hormone-responsive L2 cells. This breast tissue cell atlas provides new insights into the analysis of human breast cell systems. The atlas lays the foundation for understanding the aberrant differentiation of breast cancer cells.

Pal [84] utilized two different sequencing platforms, the 10× Genomics Chromium System 21 for large-scale analyses and the high-resolution Fluidigm C1 platform, to perform scRNA-seq. from puberty, a transcriptional transition from relatively homogeneous to heterogeneous was observed and rare subpopulations of basal cells were identified. In addition, breast basal cell hybridization profile intermediates with a propensity for luminal differentiation were identified during puberty and pregnancy. Single-cell data from different developmental stages provide a valuable resource for deciphering breast cell regulatory mechanisms at the single-cell level.

Wu [85] used scRNA-seq to map single cells from dairy tissues covering 10 different tissue types, including breast tissues, of lactating Holstein cows. A total of 88,013 high-quality single cells were obtained for downstream analysis, of which 11,436 cells were from the breast. The study focused specifically on neutrophils in the breast, and through pseudo-temporal analyses, it was determined that these neutrophils are responsive to steroid hormones during vascularization, epithelial tissue formation, and breast duct migration. These new findings suggest that neutrophils play a very important immune role in breast tissue.

The trait of dairy cows to produce milk far in excess of the nutritional requirements of their young is critical to the economic value of dairy cows. High yield, a unique production trait in dairy cows, can be effectively enhanced by traditional selection methods. As an ideal model for lactation biology research, it is important to explore the mechanistic basis of this complex trait at the cellular level. scRNA-seq of milk samples from two Holstein cows at mid-lactation (75 and 93 days, respectively) was performed by Becker [86] using the 10× Chromium platform with the aim of revealing milk transcriptome profiles at the single-cell level. In the study, milk was first cell pelleted and fat was removed by centrifugation, then 9313 and 14,544 cells were extracted from the milk samples of each cow, respectively, and these cell suspensions were loaded on different passages for sequencing, and after sequencing and subsequent filtering at the cellular and gene levels, final data were obtained for 7988 and 13,973 high-quality cells, including milk-producing cells, progenitor cells, macrophages, monocytes, dendritic cells, T cells, B cells, mast cells, and neutrophils. These findings provide valuable resources and new perspectives for a deeper understanding of the breast gland’s multiple functions, such as lactation, tissue renewal, natural immunity, protein and lipid synthesis, and hormonal responses.

Fan [87] conducted a comprehensive transcriptome analysis of breast adipocytes at different stages of porcine breast gland development using single-cell transcriptome sequencing, and identified the gene expression patterns specific to adipocytes at different developmental stages, such as puberty, pregnancy and lactation. The study not only revealed the heterogeneity of gene expression in adipocytes during development, but also emphasized the involvement of these cells in a variety of biological processes such as fat metabolism, inflammatory response and cell proliferation. By providing a detailed transcriptome profile of porcine breast adipocytes, this study greatly contributes to the understanding of the functional role of adipocytes in breast development and provides potential insights into improving porcine lactation performance in livestock production.

Basal-like and triple-negative subtypes of breast cancer are highly invasive. To investigate their origin, a deeper understanding of the heterogeneity of normal breast basal epithelial cells is required. Although the basal compartment is known to contain stem cells, a comprehensive analysis of its transcriptional composition is lacking. Guadalupe [88] systematically characterized the heterogeneity of breast basal cells using single-cell RNA sequencing, multiplexed RNA in situ hybridization, and bioinformatics methods to reveal molecular signatures and to predict the differentiation kinetics and communication patterns of the neobasal cell state. Four major transcriptional states were identified by genetic cell marker tracing, demonstrating different functional activities and metabolic preferences. In particular, the Egr2 expression state was found to be obligatory for basal cells in pubertal breast morphogenesis. This study not only provides a systematic approach to understand the heterogeneity of breast basal cells, but also reveals unknown transcriptional state dynamics.

Adult breast tissue consists of a complex network of epithelial tissue and lobular networks embedded in connective and adipose tissue. Previous studies have focused on the breast epithelial system, and many non-epithelial cell types have been understudied. Kumar [89] constructed a comprehensive human breast cell atlas (HBCA) using single-cell and spatial histology techniques. Single-cell transcriptomic analysis of 535,941 cells from 62 women and 120,024 nuclei from 20 women identified 11 major cell types and 53 cell states, revealing abundant pericyte, endothelial, and immune cell populations, as well as highly diverse luminal epithelial cells. Simultaneous spatial localization using three techniques revealed a rich and previously unrecognized ecosystem of tissue-resident immune cells within the ducts and lobules, as well as distinct molecular differences between ductal and lobular regions (Figure 1). These data provide a comprehensive reference value for the study of breast developmental biology and diseases such as breast cancer.

Initially, a large number of cell-based experiments were necessary to gain insight into the different cell populations within an organism. Until the advent of single-cell histology analysis, cell subpopulation classification and interactions remained unclear. Similarly, while it is relatively simple to observe gene expression in tissues, it is more challenging to understand gene expression in individual cells. Currently, researchers are increasingly focusing on single cells and making progress in identifying new cellular subpopulations. Single-cell RNA sequencing (scRNA-seq) is used to reveal intercellular heterogeneity, interactions between cellular subpopulations, and to study the impact on tissue and organ functions [90]. scRNA-seq plays a key role in identifying unknown cell types by transcriptional differences between cells [67]. scRNA-seq allows researchers to delve deeper into gene expression functions and mechanisms of action in rare cell types, while discovering unknown cell types [91].

The immune cells in breast also play an important role, and single-cell transcriptomic analysis of isolates from tumor tissue characterized heterogeneous tumor cells with adjacent stromal and immune cells. Chung [92] analyzed 515 breast cancer cells from 11 patients. At single-cell resolution, cancer cells exhibited common features within the tumor, whereas the majority of non-cancerous cells were immune cells consisting of three main types: the T-lymphocyte, the B-lymphocyte, and the macrophage populations. Both the T-lymphocytes and the macrophages exhibited immunosuppressive features. Of these, T lymphocytes exhibit regulatory and depletion phenotypes, whereas the macrophage population exhibits a predominantly M2 phenotype. This study showed that 16.7% of normal breast tissue contains immune cells, including four main types: myeloid, natural killer (NK), T, and B cells, located primarily in the ducts and lobules of the four breast regions of the breast. Understanding the subtle differences in immune cells can help develop effective immunotherapies that target specific cancer subtypes and determine their role in cancer. As the technology continues to improve, scRNA-seq is expected to be more widely used in the future. Single-cell sequencing provides a new technique for identifying new or rare cell types and discovering marker genes.
ijms-25-13706-t001_Table 1Table 1Breast cell type recognition marker genes and species origin.Cell TypeMarker GenesSpecies OriginLuminal hormone-responsive cells*AREG* [93], *MUCL1* [94], *AZGP1* [95], *PIP* [96],*KRT18* [97], *AGR2* [97], *ANKRD30A* [27]*Human*, *Mouse*Luminal secretory epithelial cells*SCGB2A2* [94], *SLP1* [98], *WFDC2* [99], *LTF* [100],*KRT15* [101], *MMP7* [102], *SCGB3A1* [103]*Human*, *Mouse*Basal myoepithelial cells*KTR14* [78], *KRT17* [104], *DST* [103], *KRT5* [105],*SAA1* [106], *ACTA2* [27], *SFN* [107]*Human*, *Mouse*Lymph endothelial cells*CCL21* [108], *TFF3* [109], *MMRN1* [110], *CAVIN2* [111], *CLDN5* [112], *LYVE1* [96]*Human*, *Mouse*Vascular endothelial cells*SELE* [113], *ACKR1* [114], *FABP4* [115],*ANGPT2* [116], *CSF3* [117]*Human*, *Mouse*T cells*IL7R* [118], *CCL5* [104], *PTPRC* [119], *CXCR4* [120], *GNLY* [121], *CD2* [122], *SRGN* [120]*Human*, *Mouse*B cells*IGKC* [123], *IGLC2* [124], *IGHA1* [105], *IGLC3* [105],*JCHAIN* [125], *IGHA2* [126], *IGHG1* [127]*Human*, *Mouse*Bone marrow cells*HLA-DRA* [128], *IL1B* [129], *HLA-DPA1* [130],*HLA-DPB1* [131], *HLA-DRB1* [132], *CD74* [133],*CCL3* [128]*Human*, *Mouse*Fibroblast cells*DCN* [134], *APOD* [135], *CFD* [114], *TNFAIP6* [136], *LUM* [137], *COL1A2* [134], *COL1A1* [138]*Human*, *Mouse*Perivascular cells*RGS5* [105], *IGFBP5* [139], *STEAP4* [140],*MYL9* [141], *IGFBP7* [142], *ADIRF* [143]*Human*, *Mouse*Mast cells*CMA1* [104], *CPA3* [104], *CTSG* [144],*KIT* [145], *TPSD1* [134], *tryptase* [146]*Human*, *Mouse*Adipocyte cells*APOD* [104], *CFD* [147], *DLK1* [148], *SCARA5* [104]*Human*, *Mouse*


## 4. Advances in scRNA-Seq and Combined Multi-Omics Strategies in Breast Tumor Research

Currently, scRNA-seq technology has been used in various research fields. This technology has enabled us to gain a deeper understanding of the cells and their interactions within breast tissue as well. scRNA-seq has entered a mature stage of development, but it still faces certain limitations. Differences in cell dissociation methods and tissue specificity may affect the results, and differences in the integrity of dissociated cells may also lead to differences in the results [149]. In the future, the choice of tissue or cell isolation method will remain key to constructing a single-cell atlas of the breast. Varying digestion time or enzyme temperature can help overcome problems such as disproportionate enrichment or depletion of certain cell types during sample preparation. Cell subtype identification also requires the use of multiple marker genes, rather than relying on a single marker, for accurate identification of cell subtypes within a tissue [150]. Additionally, relevant validation experiments are needed to verify it. Looking forward, the integration of multiple methods will further improve the reliability and accuracy of single-cell transcriptome sequencing studies.

The first step of scRNA-seq requires the preparation of cells in single-cell suspensions and the construction of single-cell libraries by single-cell isolation techniques. However, this process inevitably leads to the loss of cell spatial location information. Spatial heterogeneity is an important feature of tissue and organ function, and cell location information is fundamental to the study of cell development regulation and cell generation [151]. Although the scRNA-seq dataset describes relationships between cells, it is unclear where these cell populations obtained by sequencing are spatially located in the original samples, whether they are closely connected or far apart. It is also unclear how organ or tissue developmental processes are linked to specific tissue structures or distance relationships between specific cells. Although marker genes are known to be used to identify cell types, the spatial context of tissue genes is obtained by restoring the original spatial localization of cells, and the number of detected target genes is still insufficient compared to the high density of genetic information within cells [152].

To simultaneously capture cellular transcriptional heterogeneity and spatial positional information. In 2016, Joakim Lundeberg’s [153] research group introduced the concept of spatial transcriptomics and published the first in situ capture-based RNA spatial transcriptomics technology, ushering in a new era of spatial transcriptomics (ST). This technology characterizes and analyzes cell type expression profiles in a spatial dimension, allowing us to understand spatial expression differences in physiological and pathological states of tissues and organs [154]. Further, it can dissect transcripts in tissues at different spatial locations. Spatial transcriptomics can study cellular heterogeneity and localization in tissue space by combining single-cell sequencing technology, in situ technology and other related histological techniques, providing more precise research directions for developmental and disease studies [155]. The emergence of spatial transcriptomics has provided important research tools in a variety of fields, including tissue cell function, microenvironmental interactions, developmental genealogy tracking, and disease pathology [156]. Currently, spatial transcriptome sequencing technology is mainly divided into four technologies: spatial transcriptome based on in situ hybridization, spatial transcriptome based on high-throughput sequencing, spatial transcriptome based on in situ sequencing, and spatial transcriptome based on live cell labeling [157]. Among other things, 10× Genomics’ ST technology images fresh frozen tissue sections and places them on an array of RNA-bound capture probes, where the tissue is immobilized and permeabilized to release the RNA so that it can bind to adjacent capture probes, thereby capturing gene expression information [158]. Then, using the captured RNA as a template, cDNA was synthesized by reverse transcription and the library was prepared. Finally, the libraries were sequenced and the resultant data were visualized and analyzed to determine gene spatial expression levels.

Breast cancer is one of the most common malignancies and metastasis is the main cause of its deterioration. Liu [159] analyzed breast cancer patients and their metastatic axillary lymph nodes by single-cell RNA sequencing and spatial transcriptomics, and identified a diffuse population of cancer cells with high levels of oxidative phosphorylation, which were distributed at the leading edge of the tumors and exhibited a metabolic shift from glycolysis to oxidative phosphorylation, a phenomenon that was validated in multiple patient and external datasets and reveals the dynamic metabolic evolution of breast cancer cells in the early stages of lymph node metastasis.

Intracellular heterogeneity of tumors is a key feature of solid tumors and is particularly significant in breast cancer. Ryohei [160] combined spatial transcriptomics(ST) and single-cell RNA sequencing(scRNA-seq) techniques to analyze xenografts (PDXs) originating from patients with estrogen receptor-positive (ER+) breast cancer in order to reveal intratumor heterogeneity and estrogen-dependent tumor growth of the Significance. The study identified four distinct spatial functional compartments and found that proliferative cell populations were the main drivers of tumor growth, while estrogen-responsive cell populations were associated with good prognosis. In addition, hypoxia-induced and inflammation-associated compartments were associated with tumor invasiveness and treatment resistance. These findings deepen the understanding of the molecular characteristics of ER+ breast cancer and provide valuable information for developing new therapeutic strategies and improving patient prognosis.

Using single-cell RNA-seq and single-cell proteomic technologies, G Kenneth [161] constructed a high-resolution breast cell atlas covering cellular subtypes in different states. The study defined cellular subtypes within the alveolar, hormone-sensing, and basal epithelial profiles and revealed associations of these subtypes with cancer risk factors such as age, litter size, and BRCA2 germline mutations, while elucidating potential molecular regulatory mechanisms of breast epithelial cell (MEC) subtypes through organoid regulation experiments, providing a viable approach for targeted studies of breast changes in high-risk individuals. This multi-omics, single-cell approach provides a research template for exploring precancerous changes in the breast and other tissues.

Breast cancer incidence, molecular subtypes, and prognosis vary across races, and these differences stem in part from biological differences related to genetic ancestry. Poornima [162] explored the impact of genetic ancestry on the status of healthy breast epithelial and fibroblast cells by using single-cell nuclear transcriptome analyses and single-cell nuclear chromatin accessibility analyses to investigate the integration of women from diverse genetic ancestry with snATAC-seq and snRNA-seq data, revealing novel markers for multiple cell types and epithelial subtypes. In addition, the study identifies gene expression differences between vesicular and ductal epithelial cells, as well as age-related changes in signaling networks. Specifically, breast tissue from Native American women is enriched in a specific subpopulation of LASP cells, whereas fibroblasts from women of African and European ancestry exhibit a different gene expression status, and these findings provide an important resource for understanding breast biology and disease, shedding light on the biological basis of racial differences in breast cancer.

By applying large-scale single-cell and single-nucleus multi-omics techniques, as well as spatial transcriptomics and multiplex imaging, Michael [163] analyzed 61 breast tissue samples from 37 breast cancer patients to reveal characteristic associations between breast cancer subtypes and their cells of origin in terms of gene expression and chromatin accessibility, and the study determined the precise cellular origins and transcriptional networks of the breast cancer subtypes, and discovered that the importance of *BHLHE40* and *KLF5* transcription factors in the regulatory network of breast cancer, and identified key genes such as *PRKCA* and *SOX6*. In addition, this study revealed differences in immune dysfunction in breast cancer subtypes, providing new perspectives for the study of breast cancer lineage development.

To reveal the cellular subpopulation composition and spatial structure of breast phyllodes tumors (PTs), Li [164] explored their pathogenesis and diagnostic markers by scRNA-seq and spatial transcriptome analysis, and found that the tumor stromal cells contained seven subpopulations, in which the cancer-associated fibroblast-like mesenchymal cells interacted with the epithelial progenitor cells and underwent mesenchymal transition. Two stromal subpopulations express epithelial progenitor and mesenchymal markers and may differentiate into a transcriptionally active stromal subpopulation that expresses *COL4A1/2*. Binding of *COL4A1/2* to *ITGA1/B1* influences tumor growth patterns. Microproteomics studies revealed intratumor and inter-patient heterogeneity. Immunohistochemical analysis showed that *COL4A1/2* and *CSRP1* are clinically relevant for the diagnosis and grading of PTs, and this study provides new insights into the diagnosis and treatment of breast PTs.

## 5. The Prospective Outlook of scRNA-Seq Technology in Breast Development

scRNA-seq has transformed the field of developmental biology by providing unprecedented resolution of cellular heterogeneity and dynamics in complex tissues. In the context of breast development, scRNA-seq holds the promise of elucidating the molecular and cellular processes underlying normal and pathological growth. Here, we provide insights into the prospects of scRNA-seq technology, including the current and future directions of sequencing costs, computational power, and data analysis techniques. scRNA-seq’s cost has been a significant barrier to its widespread adoption. Advances in sequencing technology and library preparation methods have led to a dramatic reduction in cost, and the cost of scRNA-seq has been reduced to the point where a single laboratory can afford to conduct small- to medium-sized studies, hence the increase in the number of publications using scRNA-seq to study breast development. As costs continue to decrease, we can expect scRNA-seq to become the standard tool in the field, which will enable large-scale, population-based studies to reveal the genetic and environmental factors that influence breast development. In addition, the reduction in cost will facilitate the integration of scRNA-seq into the clinical setting for diagnostic and prognostic purposes.

scRNA-seq data analysis is computationally intensive and requires significant resources for data storage, processing and interpretation. Advances in the development of high-throughput computing based on large servers have made it possible to process large datasets generated by scRNA-seq. However, challenges remain in data sharing and standardization. The development of more efficient algorithms and the availability of computing resources will continue to improve the speed and accuracy of scRNA-seq data analysis. In the future, machine learning and artificial intelligence will play a greater role in automated cell type identification, clustering, and trajectory inference. scRNA-seq data will also become more seamlessly integrated with other histological data types, providing a more comprehensive view of breast development.

Analysis of scRNA-seq data is a complex process that requires sophisticated tools and methods. Many tools are available for preprocessing, normalization, clustering and differential expression analysis. However, the interpretation of these data remains challenging and requires more reliable and standardized methods. The field is moving towards more integrated approaches that combine scRNA-seq with other types of single-cell data (e.g., single-cell ATAC-seq or proteomics), which will give us a more comprehensive understanding of the regulatory networks controlling breast development. In addition, the development of more accurate cellular maps will facilitate cell type and state annotation. The use of spatial transcriptomics to complement scRNA-seq will also provide spatial context for the cellular heterogeneity observed in breast tissue.

The scRNA-seq technology has become a powerful tool for the study of breast development and disease, and its reduced cost and improved computational and data analysis capabilities will open new milestone in the field of breast biology research. In the future, the technology is expected to play a central role in the field of precision medicine, revealing the molecular mechanisms of breast diseases and contributing to the discovery of new therapies.

## 6. Conclusions

With the deepening of single-cell sequencing research, mysteries surrounding cell fate decisions and life processes are gradually being unraveled. By analyzing cellular genomes or transcriptomes at the single-cell level, biological research can reach unprecedented resolution and scale, representing a revolution in the breast system and other organs. scRNA-seq provides new perspectives and methods for life science research. With the continuous improvement of scRNA-seq technology and its integration with multidisciplinary techniques and algorithms, next-generation genome sequencing will usher in a new revolution, which will become an important tool for studying breast cell types and states, facilitating the development of future therapeutic strategies for breast diseases and analyzing the mechanisms of breast cell morphogenesis. 

## Figures and Tables

**Figure 1 ijms-25-13706-f001:**
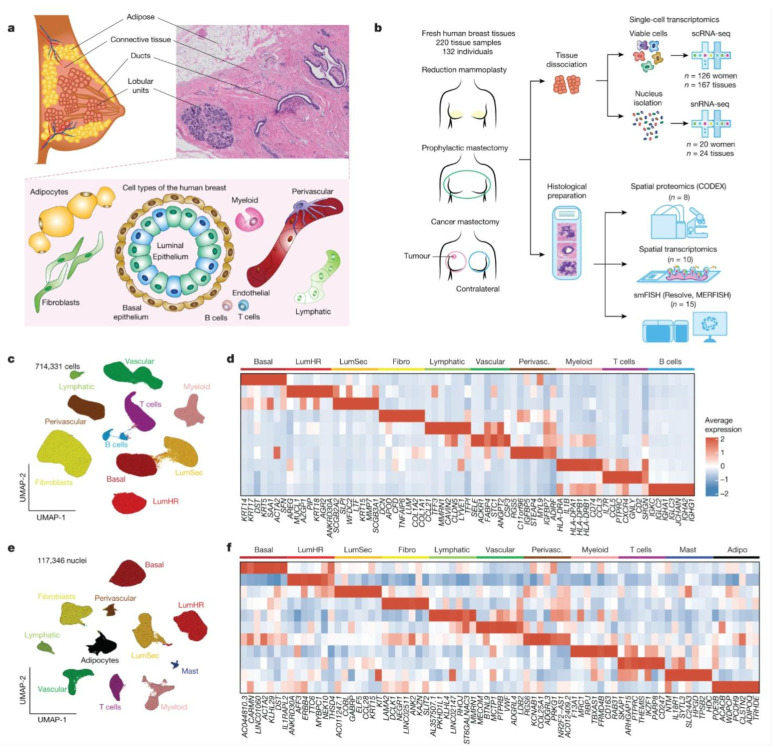
The main cell type of the adult breast. Note: Nature (2023). A spatially resolved single-cell genomic atlas of the adult human breast [89]. Retrieved from: https://www.nature.com/articles/s41586-023-06252-9. (accessed on 8 October 2024). (**a**) Anatomy of the adult human breast and a pathological hematoxylin and eosin (H&E) section, with illustrations of the major breast cell types. Scale bars, 100 μm. (**b**) The workflow of the HBCA project. (**c**) Uniform manifold approximation and projection (UMAP) of scRNA-seq data from 714,331 cells integrated across 167 tissues from 126 women, showing 10 clusters that correspond to the major cell types. (**d**) Consensus heat map of the top seven genes expressed in each cell type cluster from averaged scRNA-seq data. (**e**) UMAP representation of snRNA-seq data from 117,346 nuclei integrated across 24 tissues from 20 women, showing 11 cell type clusters. (**f**) Consensus heat map of the top seven genes expressed in each cell cluster from averaged snRNA-seq data. Adipo., adipocytes; perivasc., perivascular cells.

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
