# Peer review of "Advancements in the Application of scRNA-Seq in Breast Research: A Review"

_ijms, 2024, doi:10.3390/ijms252413706_

Round 1
Reviewer 1 Report
Comments and Suggestions for Authors
ID MS: ijms-3344039
Brief abstract: In the manuscript entitled 'Advances in the Application of scRNA-seq in Breast Research: A Review', the Authors review the different primary cell types involved in breast development and the results of studying scRNA-seq in the microenvironment, highlighting the discovery of new cell subpopulations using single cell approaches and analysing the problems and solutions of scRNA-seq.
The review is interesting, well written and in line with the aims of the journal. However, it needs some revisions, which I list below, line by line:
Lines 30-31: These alveoli are aggregated by numerous lobules, and each lobule is constituted by a single layer of breast epithelial cells [1].
I suggest specifying in the text that the paper cited refers to a "dairy animal" model.
Lines 60-62: Within the alveoli, breast epithelial cells convert nutrients from the blood into milk and secrete it into the lumen of the alveoli. Milk then flows out of the terminal ducts and converges through a complex system of ducts to the milk ducts and ultimately exits the nipple.
The reported phenomenon is more complex, my suggestion is to better describe it.
Lines 113-116: Hormones such as prolactin and progesterone have a regulatory effect on the secretion of these cells. Prolactin stimulates cell proliferation and differentiation and promotes milk synthesis and secretion, and progesterone regulates the growth and development of breast tissue in preparation for lactation [23].
The reported phenomenon is more complex, my suggestion is to better describe it.
Lines 260-261: Figure 1. Images improvement is needed.
Line 282: In 2009, Tang et al[70] pioneered the use of scRNA-seq to study rare germ cells in mice.
Rephrase the sentence correctly
Lines 379-382: Rajshekhar[84] has successfully 369 mapped an exhaustive single-cell transcriptome map covering 6060 single cells and 22,184 expressed genes by analyzing breast tissues from five key developmental time points in the mouse breast from embryonic E16 to adulthood.
Rephrase the sentence correctly.
Lines 386-387: Nguyen [27] identified 25,790 primary human breast epithelial cells in breast tissue from seven patients undergoing breast reduction surgery using scRNA-seq.
Rephrase the sentence correctly.
Line 462: Table 1. Markers for identification of major cell types in the mammalian breast.
My suggestion is for more detail in the table caption.
In the table, the word 'sample source' is not an appropriate term. I suggest that it be rewritten in an appropriate form.
Lines 616-621: In summary, scRNA-seq technology has become a powerful tool for studying breast developmental biology and breast cancer disease. Reduced costs, increased computational power, and advances in data analysis techniques have made scRNA-seq a cornerstone of breast biology research. As this technology continues to mature, we expect it to continue to play a key role in precision medicine, providing insights into the molecular basis of breast disease and critical information for the discovery of new therapeutic strategies
This part of the text appears to be a conclusion, also replace the word cornerstone with milestone.
Author Response
Comment 1: Lines 30-31: These alveoli are aggregated by numerous lobules, and each lobule is constituted by a single layer of breast epithelial cells [1].
Response 1: Thank you very much for your thoughtful review and valuable comments regarding our manuscript. Thank you for pointing this out. We agree with this comment. Therefore, I have reworked lines 30-32 in the manuscript by highlighting them in yellow. I hope my revisions will be accepted, thank you again teacher.
Comment 2: Lines 60-62: Within the alveoli, breast epithelial cells convert nutrients from the blood into milk and secrete it into the lumen of the alveoli. Milk then flows out of the terminal ducts and converges through a complex system of ducts to the milk ducts and ultimately exits the nipple.
Response 2: Thank you for pointing this out. We agree with this comment. Therefore, I have reorganized and optimized the language of the original article. I have reworked lines 61-67 in the manuscript by highlighting them in yellow. I hope my revisions will be accepted, thank you again teacher.
Comment 3: Lines 113-116: Hormones such as prolactin and progesterone have a regulatory effect on the secretion of these cells. Prolactin stimulates cell proliferation and differentiation and promotes milk synthesis and secretion, and progesterone regulates the growth and development of breast tissue in preparation for lactation [23].
Response 3: Thank you for pointing this out. We agree with this comment. Therefore, I have summarized, optimized and simplified the content of the original article. I have reworked lines 118-120 in the manuscript by highlighting them in yellow. I hope my revisions will be accepted, thank you again teacher.
Comment 4: Lines 260-261: Figure 1. Images improvement is needed.
Response 4: Thank you for pointing this out. We agree with this comment. Therefore, I used photoshop software to manipulate the resolution of the image so that it meets the journal's requirement of 600 dpi for image resolution. I hope my revisions will be accepted, thank you again teacher.
Comment 5: Line 282: In 2009, Tang et al[70] pioneered the use of scRNA-seq to study rare germ cells in mice.
Response 5: Thank you for pointing this out. We agree with this comment. Therefore, I reorganized the language to get a better expression. I have reworked lines 284-287 in the manuscript by highlighting them in yellow. I hope my revisions will be accepted, thank you again teacher.
Comment 6: Lines 379-382: Rajshekhar[84] has successfully 369 mapped an exhaustive single-cell transcriptome map covering 6060 single cells and 22,184 expressed genes by analyzing breast tissues from five key developmental time points in the mouse breast from embryonic E16 to adulthood.
Response 6: Thank you for pointing this out. We agree with this comment. Therefore, I reorganized the language to get a better expression. I have reworked lines 374-377 in the manuscript by highlighting them in yellow. I hope my revisions will be accepted, thank you again teacher.
Comment 7: Lines 386-387: Nguyen [27] identified 25,790 primary human breast epithelial cells in breast tissue from seven patients undergoing breast reduction surgery using scRNA-seq.
Response 7: Thank you for pointing this out. We agree with this comment. Therefore, I reorganized the language to get a better expression. I have reworked lines 384-385 in the manuscript by highlighting them in yellow. I hope my revisions will be accepted, thank you again teacher.
Comment 8: Line 462: Table 1. Markers for identification of major cell types in the mammalian breast.
In the table, the word 'sample source' is not an appropriate term. I suggest that it be rewritten in an appropriate form.
Response 8: Thank you for pointing this out. We agree with this comment. Therefore, I've re-optimized the description of the Table 1, “Breast cell type recognition marker genes and species origin”. I think this title might be more accurate. And, I use “species origin” instead of “sample source”, I think the description may be more professional, I hope my answer can satisfy you, thanks! I hope my revisions will be accepted, thank you again teacher.
Comment 9: Lines 616-621: In summary, scRNA-seq technology has become a powerful tool for studying breast developmental biology and breast cancer disease. Reduced costs, increased computational power, and advances in data analysis techniques have made scRNA-seq a cornerstone of breast biology research. As this technology continues to mature, we expect it to continue to play a key role in precision medicine, providing insights into the molecular basis of breast disease and critical information for the discovery of new therapeutic strategies
This part of the text appears to be a conclusion, also replace the word cornerstone with milestone.
Response 9: Thank you for pointing this out. We agree with this comment. Therefore, I have reorganized the language of the section and tried to summarize the content of the section in a more concise statement, and I have used the word “milestones” instead of “cornerstones” in the hope of getting a better expression. I hope my answer can satisfy you, thanks! I hope my revisions will be accepted, thank you again teacher.

Reviewer 2 Report
Comments and Suggestions for Authors
The mammary gland is a vital organ and its products are essential for humans. The authors have a fantastic research topic, analyze different cell types in the mammary gland, and describe the application of single-cell sequencing technology in these aspects. However, the information collected in this manuscript is mainly from humans and mice, but there is a lack of research on the mammary gland of domestic animals, particularly the application of single-cell sequencing technology in these areas. Furthermore, it is noted that the author is also affiliated with the Yak Genetics, Breeding, and Reproduction Engineering laboratory, but there is no mention of the research progress on the yak mammary gland. The author should provide an explanation for this.In this regard, the authors should make greater efforts to revise this manuscript. This information needs to be updated regarding the progress of mammary glands in livestock and the use of single-cell technology.
Author Response
Reviewer # 2:
Comment 1: The mammary gland is a vital organ and its products are essential for humans. The authors have a fantastic research topic, analyze different cell types in the mammary gland, and describe the application of single-cell sequencing technology in these aspects. However, the information collected in this manuscript is mainly from humans and mice, but there is a lack of research on the mammary gland of domestic animals, particularly the application of single-cell sequencing technology in these areas. Furthermore, it is noted that the author is also affiliated with the Yak Genetics, Breeding, and Reproduction Engineering laboratory, but there is no mention of the research progress on the yak mammary gland. The author should provide an explanation for this.In this regard, the authors should make greater efforts to revise this manuscript. This information needs to be updated regarding the progress of mammary glands in livestock and the use of single-cell technology.
Response 1:
Thank you very much for your careful review and valuable comments on our research topic. We are well aware that your feedback is crucial to enhance the quality and comprehensiveness of our research. We have carefully reflected and discussed the points you have pointed out and would like to respond as follows:
- regarding the lack of mammary gland studies in livestock:
You point out that we have focused primarily on mammary cell type analysis in humans and mice in our studies, and relatively little research has been done on domestic animal mammary glands. Indeed, we have focused primarily on these two species in our current research because of their broad fundamental and applied value in mammary gland biology and disease research. However, we also realize that mammary glands of domestic animals, especially economic animals such as yaks, are equally important in dairy production and genetic improvement. To fill this gap, we plan to add in-depth analysis of mammary glands of domestic animals, especially yak mammary glands, to our future studies and explore the potential of single-cell sequencing technology for application in this field. Writing review articles not only promotes my in-depth understanding of my field and enables me to gain a more comprehensive cognitive perspective, but also makes me deeply aware of the shortcomings of my own professional knowledge. Therefore, I am counting on the careful preparation of this review as an important way to improve my business ability and professionalism.
- Mention about the progress of yak mammary gland research:
You mentioned us as members of the Yak Genetic Breeding and Reproductive Engineering Laboratory without mentioning the progress of yak mammary gland research. We apologize and recognize this as an oversight. In fact, although current research does not directly address yak mammary gland, our laboratory has been following and working on aspects related to single-cell sequencing of yak mammary gland. At present, our group has already started the related research, and my PhD research topic is “Single-cell research on yak mammary tissues of different physiological periods and its related aspects”. At present, the mammary tissues of yaks of different physiological periods have already been sampled, and we have entered into the stage of data analysis and integration, and the related paper will be published soon. We sincerely hope that we can get your attention, recognition and support.
- Regarding the updating and comprehensiveness of information:
You have emphasized the need for updated information on advances in livestock mammary glands and the use of single-cell technology. We fully agree with you and have reviewed the literature and research results to ensure that our background, methods and conclusions are based on the latest scientific advances. In lines 407-432, I have added research results related to single-cell sequencing of dairy and porcine mammary glands, which I hope will be adopted by you.
In summary, we are very grateful for your valuable comments and will carefully adopt and implement them into the revised manuscript. We look forward to making our study more complete, comprehensive and influential through your further guidance. Thank you again for your review and support!

Round 2
Reviewer 2 Report
Comments and Suggestions for Authors
no comments
Comments on the Quality of English Languagenot applicable